# Verification Method of Safety Properties of Embedded Assembly Program by Combining SMT-Based Bounded Model Checking and Reduction of Interrupt Handler Executions

**Satoshi Yamane *,†,‡, Junpei Kobashi † and Kosuke Uemura †**

Graduate School of Natural Science and Technology, Kanazawa University, Kanazawa 920-1192, Japan;
jkb.recruit256@gmail.com (J.K.); kuemura@csl.ec.t.kanazawa-u.ac.jp (K.U.)

* Correspondence: syamane@is.t.kanazawa-u.ac.jp; Tel.: +81-76-234-4856
† Current address: Kakumamachi, Kanazawa City.
‡ The author contributed to this work.

**Abstract:** Embedded software has properties dependent on hardware (direct operation of address spaces, memory mapped I/O, interruption, etc.). Therefore, demands about the established method of formal verifications corresponding to those properties are increasing from the point of view of shorter development and high reliability. Our study aims at enabling a formal verification with Satisfiability Modulo Theories-Based Bounded Model Checking (SMT-Based BMC) of safety for embedded assembly codes. Our proposed method generates models of assembly codes in detail with the fixed-sized bit-vectors theory. The models generated by our method include interrupts, and the size of the models is reduced using Interrupt Handler Execution Reduction (IHER) technique. In this paper, we have developed the verification method of safety properties of embedded assembly program by combining SMT-Based Bounded Model Checking and Reduction of Interrupt Handler Executions. Moreover, we show the evaluation of our method by experiments using prototype model checker.

**Keywords:** embedded assembly program; SMT; bounded model checking; safety; reduction of interrupt handler executions

## 1. Introduction

### 1.1. Background

Embedded systems are widely used, and the complexity of hardware and software advances with many functionalization. Also, embedded systems similar to those in Reference [1] based on microcontrollers, which are employed in airplanes, cars and mobile phones, have been targeted. The software of embedded systems has to be tested extensively because errors such as stack overflows and exception handlings may lead to severe or even fatal events as loss of reputation as in the case of the Toyota Prius bug [2]. The embedded software is mostly written in C or in assembly language. Removing errors in embedded software is difficult because deploying the updates is complicated and cost intensive. Testing of microcontroller software is often not possible because it is too time consuming for the desired time-to-market or too expensive. Also, testing is not sufficient for embedded systems as there are errors that are difficult to find by testing. There are standards such as IEC61508 [3] that recommend the application of formal methods such as formal verification methods. Formal verification methods, especially model checking [4] is very promising. In this paper, we focus on model checking.

Verifying C codes was not successful for embedded systems because the existing C code model checkers aim at the verification of hardware-independent ANSI C code [5]. In order to verify embedded systems, we must verify more features such as hardware-dependent constructs and interrupt handlers. All these features are not handled by the existing C code model checkers [5].

Verifying assembly codes is required. For example, we explain interruptions as follows. As one statement of C code supports plural instructions of assembly code, we can not verify interruptions because we cannot treat the timing when the interrupt enters well. When we verify assembly code with interruptions, the state space explosion problem occurs as each block of Control Flow Graph (CFG) [6,7] consists of one instruction. There is a possibility that an interrupt may be generated for each assembly instruction. Since there is a call to the interrupt processing routine for each instruction of the assembly, a state explosion occurs. In order to reduce the number of interruptions, B. Schlich et al. developed Interrupt Handler Execution Reduction (IHER) [8]. B. Schlich et al. have developed IHER, which reduces the number of Interrupt Handler (IH) executions by blocking IHs at program locations where there is no dependency between certain IHs and the program. There is a dependency if either one influences the other or the visible behavior of the program is changed. In order to reduce the number of blocks, we propose the method of making the block of codes, at which interruptions do not occur. We call the block Assembly Code Block (ACB).

Despite the high importance of assembly code verification, studies on verifying assembly codes, other than B. Schlich's study, almost do not exist [1]. Also, we show the advantage of the verification of assembly program that B. Schlich pointed out as follows [1].

1.  All errors (e.g., stack overflows, arithmetic overflows, interrupt handling errors, and writing reserved registers) introduced during the complete development process are consequently included in assembly program outcome, but by analyzing only the assembly program it is not possible to find and detect the roots of the assembly outcome.
2.  Assembly language usually has a clean and well-documented semantics. Vendors of microcontrollers provide documentation describing the semantics of the provided assembly constructs.
3.  Recently, bounded model checking of software using SMT solvers attracts attention [9]. SMT Solvers check the satisfiability of first-order formulas containing operations from various theories such as the Booleans, bit-vectors, arithmetic, arrays, and recursive datatypes. When model checking assembly program using SMT, the model checker does not have to exploit the compiler behavior, hardware-dependent constructs can be handled, and the source code (C code) of the software is not required. Hence, even programs that use libraries not available in source code can be analyzed. On the other hand, L.C. Cordeiro et al. have developed bounded model checking of C program using SMT [10], but bounded model checking of the assembly program using SMT has not existed.
4.  Programs consisting of components written in different programming languages can be verified. When model checking the source code, only single components can be verified, and for each programming language used, a specific model checker has to be utilized.

*1.2. Outline of This Paper*

In this paper, we develop the verification method of safety properties by combining SMT-Based Bounded Model Checking [9] and Reduction of Interrupt Handler Executions [8].

We model registers and values of assembly codes using Fixed-Size Bit-Vector theory, and construct a transition system. Finally we verify the transition system using SMT-Based Bounded Model Checking. Also we construct the transition system including interrupts. We reduce state spaces using IHER [8]. IHER reduces the number of Interrupt Handler executions by blocking Interrupt Handlers at program locations where there is no dependency between certain Interrupt Handlers and the program [8]. We propose Assembly Code Block (ACB) by extending Basic Block of Control Flow Graph (CFG) [6,7] using IHER [8] in order to reduce state spaces. Moreover we show our proposed method effective by implementing the prototype. To the best of our knowledge, there have been no

previous studies of SMT-Based Bounded Model Checking of embedded assembly program including interrupts. The outline of this research has been published at a domestic conference [11] and an international conference [12]. This paper is an extension of the previous work in References [11,12].

*1.3. Related Work*

Our paper is related to both SMT model checking of assembly program and reduction of interruption handlings. In particular, this paper is based on the model checking of assembly program [1,13], SMT-Based Bounded Model Checking [9] and Interrupt Handler Execution Reduction [8]. Therefore, this subsection presents related work regarding assembly program model checking first and then related work regarding SMT model checking and reduction of interruption handlings.

1.  B. Schlich studied a new approach to model checking software for microcontrollers, which verifies the assembly code of the software [1]. The state space is built using a tailored simulator based on static analysis, which abstracts from time, handles nondeterminism, and creates an overapproximation of the behavior shown by the real microcontroller. On the other hand, we have developed model checker by both static analysis and dynamic program analysis such as undefined values [13].

    This paper and Reference [1] use different approaches as follows:

    (a)  In Reference [1], if the model checker requests successors of a state which are not created yet, the state space uses the simulator to create the successors on-the-fly.
    (b)  On the other hand, in this paper, the verification system completes the state transition system and passes it to the SMT solver. By using SMT's background theory such as bit vector, bit-level assembly instructions can be easily expressed as functions of background theory and can be directly input to the SMT solver for verification.

2.  B. Schlich et al. studied IHER in order to reduce the number of Interrupt Handler executions [8]. Also B. Schlich et al. studied various abstraction techniques based on static program analysis [14]. We propose Assembly Code Block (ACB) by extending Basic Block of CFG [6,7] using IHER [8].
3.  L.C. Cordeiro et al. studied SMT-Based Bounded Model Checking of embedded ANSI-C program [10]. On the other hand, we study SMT-Based Bounded Model Checking of embedded assembly program. The background theory of the bit vector is effective for assembly program. Similar to model checking of C program, the background theory of the bit vector is expected to be effective in model checking of assembly program. In this study, we use the background theory of the bit vector for the assembly program, and construct the state space by representing register as a data type of a fixed length bit vector and representing the address space as a function of fixed bit vector mapping from address to value.
4.  Recently, L. Lihao et al. proposed effective verification of low-level embedded C software with nested interrupts based on both partial-order encoding and symbolic execution [15]. On the other hand, in this paper, we merge basic blocks generated by interrupt processings based on reducing the number of interrupt handlers by IHER, and verify assembly program based on SMT Bounded model checking. If the method of Lihao Liang et al. is adopted, nested interrupts can be effectively handled in our paper as well.

*1.4. The History of Model Checking*

Model checking methods have been categorized into four categories such as (1) classical model checking, (2) Binary Decision Diagram (BDD) based symbolic model checking, (3) SATisfiability problem (SAT) based symbol model checking, and (4) SMT based model checking. Also, model checking methods have evolved from (1) to (4).

1.  First, the evolution from (1) to (2) is in Reference [16]. The system is encoded in BDD without implementing the system directly in the adjacency list and so forth, realizing the verification of the large scale system.
2.  Second, the evolution from (2) to (3) is in Reference [17]. We avoid the exponential state explosion of BDD, and encode the system into propositional logic rather than BDD to realize SAT verification with bounded model checking.
3.  Finally, the evolution from (3) to (4) is in Reference [9]. SMT model checking is realized by expanding the scope of the verification by using the background theory by encoding the system into the predicate logic expression without the quantification symbol, not the propositional logic. Furthermore, SMT model checking can use a general-purpose SMT solver [18], and there is also future development. Our study is based on SMT model checking.

In this paper, in the scope of our survey, it is the first study on SMT based symbolic model checking with bounded model checking using the background theory of the bit vectors of assembly program.

*1.5. Structure of This Paper*

The rest of the paper is organized as follows. We first give a brief introduction to SMT-Based Bounded Model Checking, and in Section 3 we present SMT-Based Bounded Model Checking of assembly program. We next give a brief introduction to IHER, and in Section 5 we present ACB by extending Basic Block of CFG using IHER. Finally, we present the prototype model checker and the experiment.

## 2. Introduction to SMT-Based Bounded Model Checking

*2.1. Overview*

SMT generalizes boolean satisfiability (SAT) by adding equality reasoning, arithmetic, fixed-size bit-vectors, arrays, quantifiers, and other useful first-order theories [18]. A SMT solver is a tool for deciding the satisfiability of formulas in these theories.

On the other hand, model checking verifies whether a model $M$ satisfies a property $\Phi$ or not. SMT solvers enable SMT-Based Bounded Model Checking over infinite domains. The idea of Bounded Model Checking is to check the negation of a given property at a given depth [19].

*2.2. Program Verification by SMT-Based Bounded Model Checking*

In BMC, the program to be analyzed is modelled as a state transition system, which is extracted from the control-flow graph (CFG) [6,7] as shown in Figure 1. In Figure 1, $l_0, l_1$ and $l_2$ are a set of states, $T_0, T_1$ and $T_2$ are a set of transition relations such as $T_0 = \{(l_0, l_1)\}$, $T_1 = \{(l_1, l_1)\}$ and $T_2 = \{(l_1, l_2)\}$. This graph is built as part of a translation process from program text to single static assignment (SSA) form. A node in the CFG represents either a (non-) deterministic assignment or a conditional statement, while an edge in the CFG represents a possible change in the program's control location.

A state transition system $M = (S, T, S_0)$ is a machine that consists of a set of states $S$ of basic blocks, where $S_0 \subseteq S$ represents the set of initial states, and $T \subseteq S \times S$ is the transition relation, that is, pairs of states specifying how the system can move from state to state. A state $s \in S$ consists of the value of the program counter $PC$ and the values of all program variables of a basic block. An initial state $s_0$ assigns the initial program location of the CFG to the $PC$. We identify each transition $\gamma = (s_i, s_{i+1}) \in T$ between two states $s_i$ and $s_{i+1}$ with a logical formula $\gamma(s_i, s_{i+1})$ that captures the constraints on the corresponding values of the program counter and the program variables.

Given a transition system $M$, a property $\Phi$, and a bound $k$, BMC unrolls the system $k$ times and translates it into a VC (Verification Condition) $\psi^k$ such that $\psi^k$ can be satisfied if and only if for some $i \leq k$ there exists a reachable state at time step $i$ in which $\Phi$ is violated. The VC is a quantifier-free formula in a decidable subset of first-order logic, which is then checked for satisfiability by an SMT

solver. In this work, we are interested in checking safety properties of single-threaded programs. The associated model checking problem is formulated by constructing the following logical formula:

$$\psi^k = I(s_0) \wedge \bigvee_{i=0}^{k} \bigwedge_{j=0}^{i-1} \gamma(s_j, s_{j+1}) \wedge \Phi(s_i). \tag{1}$$

Here, $\Phi$ is a safety property, $I$ the set of initial states of $M$ and $\gamma(s_j, s_{j+1})$ the transition relation of $M$ between time steps $j$ and $j+1$ of basic blocks. Hence, $I(s_0) \wedge \bigwedge_{j=0}^{i-1} \gamma(s_j, s_{j+1})$ represents the executions of $M$ of length $i$ and Equation (1) can be satisfied if and only if for some $i \leq k$ there exists a reachable state at time step $i$ in which $\Phi$ is violated. If Equation (1) is satisfiable, then the SMT solver provides a satisfying assignment, from which we can extract the values of the program variables to construct a counterexample. A counterexample for a property $\Phi$ is a sequence of states $s_0, s_1, \ldots, s_k$ with $s_0 \in S_0, s_k \in S$, and $\gamma(s_i, s_{i+1})$ for $0 \leq i < k$. If Equation (1) is unsatisfiable, we can conclude that no error state is reachable in $k$ steps or less [10].

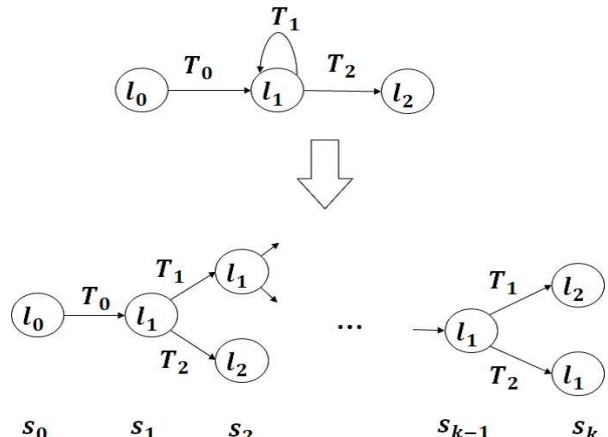

**Figure 1.** Unrolling the system from Control Flow Graph (CFG).

## 3. Proposal of Modeling Assembly Codes

We construct logical formulas by modeling assembly codes using SMT (Satisfiability Modulo Theories).

### 3.1. Defining States of Assembly Codes

A state is defined by values of registers and memory in microcontroller, and the values are represented as Bit-Vector.

In this paper, we model values of 16-bit registers and memory assigned address in the 16-bit by Fixed-Size Bit-Vector theory in SMT [18]. As some assembly instruction operates bits, Fixed-Size Bit-Vector theory is effective. We define registers by Fixed-Size Bit-Vector typed variables, and define memory by the function of Fixed-Size Bit-Vector mapping a 16-bit address to a 8-bit value. We construct states by Fixed-Size Bit-Vector typed variables and the function. Fixed-Size Bit-Vector theory consists of functions and predicates of Fixed-Size Bit-Vector. Functions consist of concat, extract, bvadd, bvsub, where function concat concatenates two bit vectors, function extract extracts bits, function bvadd adds two bits, function bvsub subtracts bits from bits. Fixed-Size Bit-Vector theory is defined by SMT-LIB 2.0, which is an international initiative aimed at facilitating research and development in Satisfiability Modulo Theories (SMT) [20,21], and is implemented by SMT solvers.

### 3.2. Defining State Transitions of Assembly Codes

First as IH might be invoked at every assembly instruction, we represent an assembly program by state transitions via Control Flow Automata (CFA) with the idea of Berkeley Lazy Abstraction

Software verification Tool (BLAST) [22,23] instead of CFG. A CFA is a directed graph, with locations corresponding to control points of the assembly program, and edges corresponding to program operations. The transition relation $\gamma(s^{CFA}_j, s^{CFA}_{j+1})$ is defined by $G(s^{CFA}_j) \wedge IS(s^{CFA}_j, s^{CFA}_{j+1})$ as shown in Figure 2, where $s^{CFA}_j$ $(j = 0, 1, 2, )$ is a state generated by one assembly instruction based on CFA. If there is a branch instruction in $s^{CFA}_j$, the state of the transition destination is $s^{CFA}_{j+1}$. In this case, the transition relation is defined by the same $\gamma(s^{CFA}_j, s^{CFA}_{j+1})$. The case of ACB (in Section 5.3) can be defined similarly.

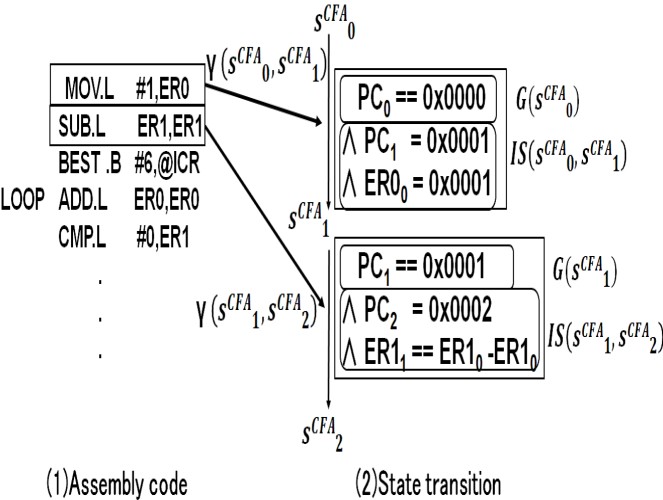

**Figure 2.** State transition of assembly code.

In this paper, we use standard Fixed-Size Bit-Vector theory [18]. In the description of CFGs, we use two kinds of equation symbols for clarity, $==$ for guard conditions and $=$ for assignments. When passing formulas to the SMT solver, both symbols are translated as the equation symbol of the Bit-Vector theory.

By the way, the transition relation $\gamma(s_j, s_{j+1})$ in Equation (1) is generated from CFG. On the other hand, the transition relation $\gamma(s^{CFA}_j, s^{CFA}_{j+1})$ is generated from CFA. $\gamma(s^{CFA}_j, s^{CFA}_{j+1})$ is different from $\gamma(s_j, s_{j+1})$.

1.  First a transition condition $G(s^{CFA}_j)$ is the equation using *PC* (Program Counter). For $PC_0 == 0x0000$, if the $PC_0$ is $0x0000$, this means that the instructions of the Static Single Assignment form (SSA) format are executed. $PC_1 = 0x0001$ means to assign $0x0001$ to $PC_1$.
2.  Next after executing assembly codes at $s^{CFA}_j$, the transition from $s^{CFA}_j$ to $s^{CFA}_{j+1}$ occurs. The next state $s^{CFA}_{j+1}$ is defined by a instruction constraint $IS(s^{CFA}_j, s^{CFA}_{j+1})$. CFA is constructed from assembly codes. Each node consists of assembly code, and each edge is caused by a instruction constraint $IS(s^{CFA}_j, s^{CFA}_{j+1})$. An instruction constraint $IS(s^{CFA}_j, s^{CFA}_{j+1})$ is represented by predicate formulas. There are instructions such as data transfer, arithmetic operation, extraction and concatenation [24]. The logical formula of state transition $\gamma(s^{CFA}_j, s^{CFA}_{j+1})$ is represented by a instruction constraint $IS(s^{CFA}_j, s^{CFA}_{j+1})$. All instructions can be transformed into $IS(s^{CFA}_j, s^{CFA}_{j+1})$.

Also, in Figure 2, for all registers that are not assigned in a state (for example, ER), a constraint such as $ER_{p+1} = ER_p$ is set to $IS(s^{CFA}_j, s^{CFA}_{j+1})$, where $p \in \{0, 1, 2, \ldots\}$. All variables that appear in the left-hand sides of SSA are distinguished by introducing a fresh variable for each assignment. We use the subscript $p$ to represent these variables. $p$ is used for variables with distinct names.

Here we show three $IS(s^{CFA}_j, s^{CFA}_{j+1})$ of *MOV* and *ADD* instructions as follows.

1.  The instruction constraint $IS(s^{CFA}{}_j, s^{CFA}{}_{j+1})$ of data transfer instruction $MOV\ ERs, ERd$ is represented by the equation between $ERs$ and $ERd$ as follows.

$$ERd_{p+1} = ERs_p \tag{2}$$

where $ERs_p$ represents the value of $ERs$ at $s^{CFA}{}_j$, and $ERd_{p+1}$ represents the value of $ERs$ at $s^{CFA}{}_{j+1}$.

2.  The instruction constraint $IS(s^{CFA}{}_j, s^{CFA}{}_{j+1})$ of Fixed-Size Bit-Vector operation, for example $ADD$ $ERs, ERd$ is represented by the equation between an operation result and a destination register. The following Equation (3) represents adding $ERs$ and $ERd$, and storing it in $ERd$ using *bvadd*.

$$ERd_{p+1} = bvadd\ ERd_p\ ERs_p. \tag{3}$$

3.  The instructions (a) $MOV\ @AR, ERd$ and (b) $MOV\ ERs, @AR$ are represented by functions *_extract* and *concat* of bits as follows. Here @ denotes register indirect addressing. Access to address space is defined by a function AS and the argument of the function is indirect address.

    (a)  The instruction constraint $IS(s^{CFA}{}_j, s^{CFA}{}_{j+1})$ of the instruction $MOV\ @AR, ERd$ is represented by Equation (4) using *concat* as shown in Figure 3. The value of 16-bit register $AR$ is interpreted as memory address, and the value that is stored at the address is stored into $ERd$.

$$ERd_{p+1} = (concat\ AS_p\ (AR_p)$$
$$AS_p\ (bvadd\ AR_p\ \sharp x0001)). \tag{4}$$

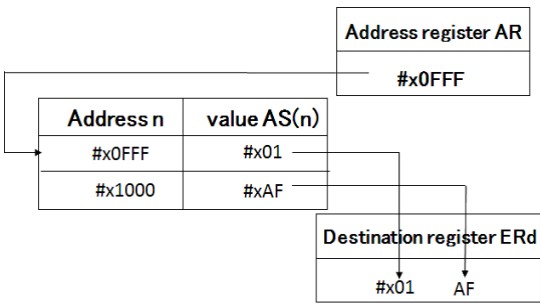

**Figure 3.** $MOV\ @AR, ERd$.

    (b)  The instruction constraint $IS(s^{CFA}{}_j, s^{CFA}{}_{j+1})$ of the instruction $MOV\ ERs, @AR$ is represented by Equation (5) using *_extract* as shown in Figure 4. The value of 16-bit register $AR$ is interpreted as memory address, and the value of $ERs$ is stored into the address $AR$.

$$( \ (AS_{p+1}\ (AR_p))$$
$$= ((\_extract\ 15\ 8)\ ERs_p)$$
$$\wedge \tag{5}$$
$$((AS_{p+1}\ (bvadd\ AR_p\ \sharp x0001))$$
$$= ((\_extract\ 7\ 0)\ ERs_p)).$$

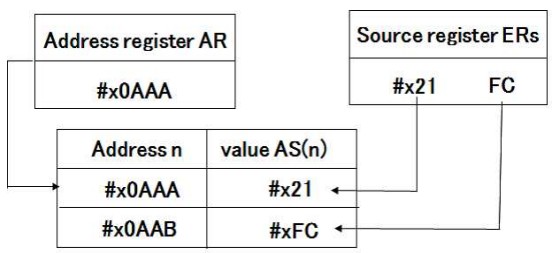

**Figure 4.** *MOV ERs, @AR.*

Because all including the update of *PC* and status registers may not be stated clearly for an operand, we add update of the *PC* and the status registers to the instruction constraint $IS(s^{CFA}_j, s^{CFA}_{j+1})$ in consideration of both the structure of the microcomputer and the syntax of assembly code.

Also in consideration of interrupts, we must compose program and Interrupt Handler (IH). Composing program and Interrupt Handler may cause the state space explosion. But in the case of a routine controlling permission and prohibition of interrupts, we need not add interrupts into the program location when interrupts are prohibited. Therefore we can reduce the state space of program using B. Schlich et al.'s method [8].

## 4. Introduction to Interrupt Handler Execution Reduction (IHER)

B. Schlich et al. have developed an abstraction technique called IHER, which reduces the number of IH executions by blocking IHs at program locations where there is no dependency between certain IHs and the program [8]. Here there is a dependency if either one influences the other or the visible behavior of the program is changed. An IH influences a program location if it writes a memory location that is accessed by the program location, and on the other hand, a program location influences an IH if it enables or disables interrupts.

In this paper, we reduce the number of IH executions using B. Schlich et al.'s IHER.

B. Schlich et al.'s IHER consists of four steps. According to B. Schlich et al.'s paper [8], we briefly explain the four steps as follows.

1. Detect Dependencies between IHs:

   $i, j \in$ IH depend on each other, if one of the following conditions holds:

   - one enables or disables the other,
   - one writes a memory location accessed by the other, or
   - one writes a memory location. Writing a memory location or a register is an atomic proposition, where an atomic proposition is appeared in temporal logic formulas. In other words, writing a value to a variable is used as the atomic proposition in a temporal logic formula that describes a property to be verified.

2. Detect Dependencies between Program and IHs:

   To detect the dependencies between the program and the IHs, we mark specific program locations with the following two labels: *execution* and *barrier*. The label *execution* implies that there exists a dependency between the preceding program location and an IH, and thus, this IH needs to be executed eventually. The label *barrier* denotes that there exists a dependency between that program location and an IH, and therefore, this IH needs to be executed before the instruction at that location is executed.

   Let program location $k$ be a direct predecessor of program location $l$.

- Label *execution*:

  For each $i \in$ IH, $l$ is labeled with *execution$_i$* if one of the following conditions is satisfied:

  - $k$ enables or disables $i$,
  - $k$ writes a memory location that is accessed by $i$, or
  - $k$ writes a memory location that is used in an atomic proposition.

- Label *barrier*:

  For each $i \in$ IH, a program location $l$ is labeled with *barrier$_i$* if one of the following conditions holds:

  - $i$ writes a memory location that is accessed by $l$,
  - $l$ enables or disables $i$,
  - $l$ writes a memory location that is accessed by $i$, or
  - $l$ writes a memory location that is used in an atomic proposition.

3. Refine Labelings:

   In the refinement step, we try to reduce state spaces further by moving *execution$_i$* labels until their execution is actually required. We move labels *execution$_i$* forward until one of the following conditions holds:

   - a program location labeled with *barrier$_i$* is reached,
   - a loop entry is found, or
   - a loop exit is found.

4. Label Blocking Locations:

   In the last step, all program locations are labeled with IHs that can be blocked at the corresponding program location. An IH can be blocked at a program location if its execution is not required. Thus, a program location is labeled with *blocking$_i$* if it is not labeled with *execution$_i$*.

IHER reduces the number of program locations at which the possible execution of interrupt handlers (IHs) has to be considered. In this paper, we reduce state spaces using IHER. B. Schilich's paper proves that model checking of $CTL^* - X$ is valid, even if IHER reduces the number of program locations at which the possible execution of interrupt handlers (IHs) has to be considered. Here $CTL^* - X$ is defined by disallowing the nexttime operator $X$ in $CTL^*$ formulas [25]. In this paper, as we verify assembly codes using subclass of $CTL^* - X$ such as safety properties, our verifications are valid even if IHER reduces the number of assembly program locations.

## 5. Proposal of Assembly Code Block (ACB) of CFG Using IHER (in Other Words, Combining SMT-Based Bounded Model Checking and Reduction of Interrupt Handler Executions)

### 5.1. Introduction to ACB

In this paper, we reduce the number of IH executions by blocking IHs at program locations using IHER, and make a set of basic blocks from assembly codes blocking IHs. We propose this set of blocks, what we call ACB. By the same method as partitioning program codes into basic blocks [6,7], we partition assembly codes into blocks. Therefore there are no jumps into the middle of the block. Also control will leave the block without halting or branching, except possibly at the last instruction in the block. The blocks become the nodes of CFG, whose edges indicate which blocks can follow which other blocks.

The start point satisfies one of the following conditions:

- A start node of program

- A branch destination node
- An *execution* node

Also the end point satisfies one of the following conditions:

- An end node of program, which has return instruction
- A node, which has branch instruction

The consecutive nodes from the start point of the block to the end point becomes one node.

### 5.2. ACB (Assembly Code Block)

We propose ACB by extending basic blocks [6,7]. In ACB, assembly codes, at which IHs are blocked, are collected, and one block is made. ACB is built as part of a translation process from assembly program text to SSA (static single assignment) form [6,7] as shown in Figure 5, where $s_j$ is generated by a basic block of CFG, and $s^{ACB}{}_q$ is generated by ACB of CFG.

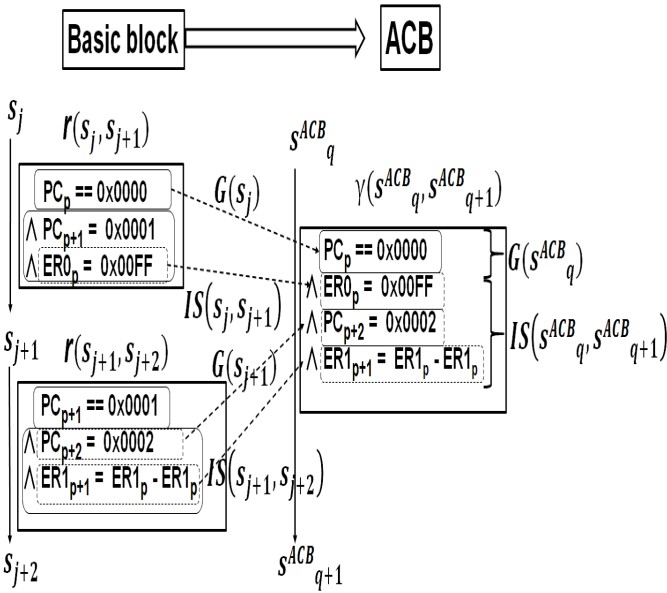

**Figure 5.** Transition of Assembly Code Block (ACB).

Basic blocks $\gamma(s_{j+m}, s_{j+m+1}) = G(s_{j+m}) \wedge IS(s_{j+m}, s_{j+m+1})$ $(m = 0, \ldots, L-1)$ is defined by transforming assembly program into SSA form and then into the quantifier-free formula, where $L$ is the number of instructions merged from basic blocks to ACB in program including interrupt routines. ACB based on IHER reduces the number of logical formulas of a transition $\gamma(s^{ACB}{}_q, s^{ACB}{}_{q+1})$, where $\gamma(s^{ACB}{}_q, s^{ACB}{}_{q+1}) = G(s^{ACB}{}_q) \wedge IS(s^{ACB}{}_q, s^{ACB}{}_{q+1})$ as shown in Figure 5, where $q \in \{0, 1, 2, \ldots\}$ represents the order in which $s^{ACB}{}_q$ is generated. $s^{ACB}{}_q$ consists of the values of variables of the bit vector type representing the values of the program counter, program variables and registers. Each $s^{ACB}{}_q$ is defined by the values of common state variables. If no assignment is made to a register, an expression is added to indicate that the value of $ER0_{p+1} = ER0_p$ in state $s_{j+1}$ does not change. But we omit the expression in Figure 5 and all figures in this paper. In Figure 5, we denote variables such as $PC_p$ and $ER1_{p+1}$ by SSA notation. Figure 5 shows a method of constructing $ACB(s^{ACB}{}_q, s^{ACB}{}_{q+1})$ from the basic blocks $\gamma(s_j, s_{j+1})$ and $\gamma(s_{j+1}, s_{j+2})$. Here $PC_p == 0x0000$ means that if $PC_p$ is equal to $0x0000$, the following instructions will be executed, and $PC_{p+1} = 0x0000$ means to substitute $0x0000$ into $PC_{p+1}$. $PC_p$ and $PC_{p+1}$ are SSA forms of a single state variable $PC$, and $PC_p$ and $PC_{p+1}$ correspond to a state variable $PC$. In Figure 5, each set of variables used in $s_j, s_{j+1}, s_{j+2}$ and $s^{ACB}{}_q, s^{ACB}{}_{q+1}$ is the following subset:

$$\{PC_p, PC_{p+1}, ER0_p, PC_{p+2}, ER1_p, ER1_{p+1}\}$$

1. $G(s^{ACB}{}_q)$ is the conditional statement using *PC* value of the first instruction.
2. $IS(s^{ACB}{}_q, s^{ACB}{}_{q+1})$ is the conjunction of logical formulas representing update of registers and locations by executing instructions. As shown in Figure 5, the number of logical formulas of ACB is smaller than the number of logical formulas of basic block.

The generation procedure of ACB formula such as $\gamma(s^{ACB}{}_q, s^{ACB}{}_{q+1}) = G(s^{ACB}{}_q) \wedge IS(s^{ACB}{}_q, s^{ACB}{}_{q+1})$, is as follows.

**Description 1.**

*(1)* *Since the first instruction of ACB is a guard instruction, a predicate logical expression $G(s^{ACB}{}_q)$ with a conditional expression == is generated.*

*(2)* *Since the instruction after the first is an assignment statement to a register or a memory, a logical expression is generated in the usual SSA format [7].*

*(3)* *Repeat (2) to generate these logical products.*

*(4)* *Finally generate a logical product of (1) and (3).*

Here we describe "the condition of a basic block" [6,7] as follows. A basic block is a sequence of instructions, satisfying "the condition of a basic block" such that the flow of control can enter the basic block through the first instruction and control will leave the basic block without halting or branching except at the last instruction in the basic block.

Finally we define Algorithm 1 to generate from basic blocks to ACB.

---

**Algorithm 1** Algorithm to generate from CFA to ACB

---

1: **Input** : CFA of assembly program
2: **Output** : CFG of ACB
3: **Method** : By the following procedures of (1), (2) and (3), ACB is generated from assembly program.

**Description 2.**

*(1)* *First, as IH might be invoked at every instruction and IH is called for all instructions in CFA [22,23], we embed IH call in every instruction.*

*Next, as shown in Figure 6, for all instructions, by making both an edge from IH to an instruction and an edge from the instruction to IH, we generate CFG using commonly used techniques in compiler [6,7].*

*(2)* *First, as shown in Figure 7, we delete unnecessary IH calls using IHER [8].*

*Next, after deleting unnecessary IH calls, we reconfigure CFG for basic blocks.*

*(3)* *After IHER removes unnecessary IH calls, there are some basic blocks where IH branches are deleted.*

*The basic blocks where the branches are deleted can be merged, so we merge and configure a new basic block, what is called ACB as shown in Figure 8, as follows:*

*Generating from basic blocks to ACB is performed by depth-first search from the basic block of the CFG root as follows:*

**Description 3.**

*(a)* *The following processing is performed on the next basic block connected by an edge from the basic block:*

**Description 4.**

*(a)-1* *Merge basic blocks if "the condition of a basic block" is satisfied. However, we do not merge with basic blocks that have already been merged with other basic blocks. Two ACBs can meet the same block when merging and bifurcation exist in the CFG.*

*(a)-2* *Otherwise, do not merge, and the basic block becomes ACB.*

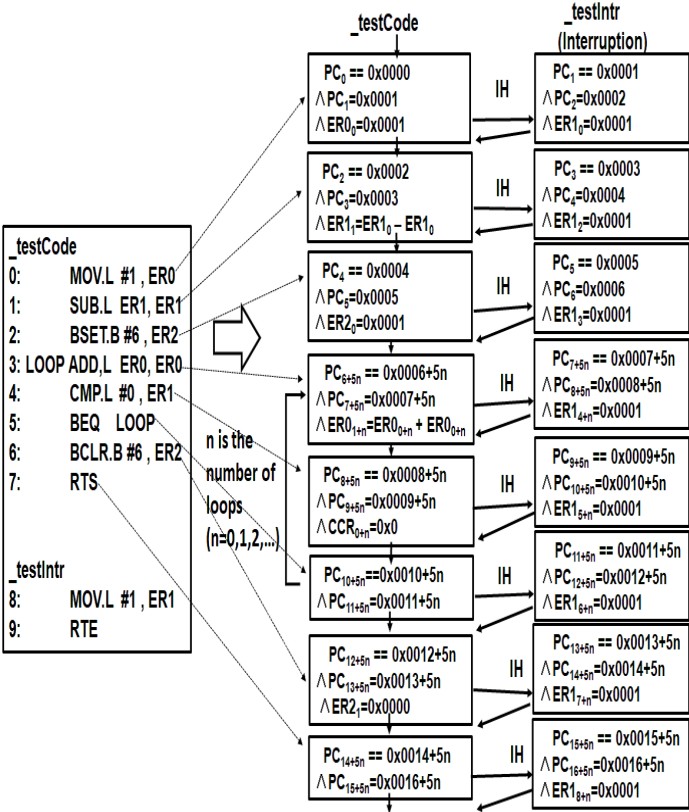

**Figure 6.** Usual CFG.

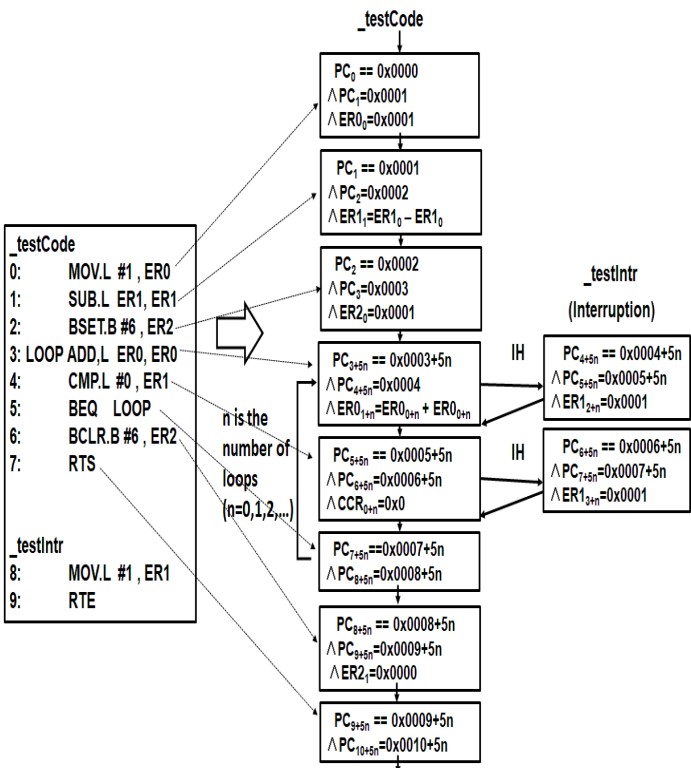

**Figure 7.** CFG of IHER.

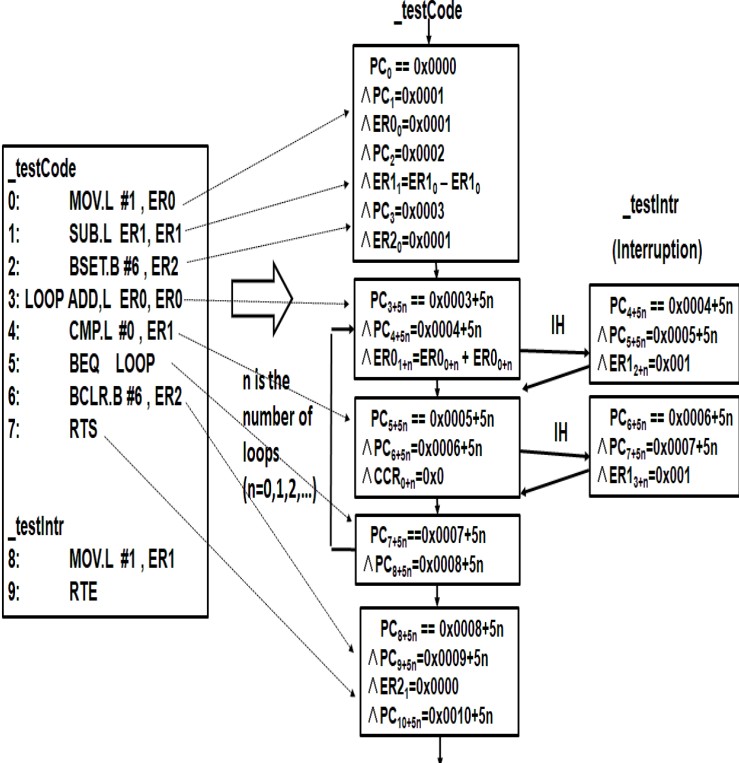

**Figure 8.** CFG of ACB.

### 5.3. Program Verification Based on ACB

In order to verify program using SMT-Based Bounded Model Checking based on ACB, first the assembly program is modeled as a state transition system, which is extracted from CFG consisting of ACBs, instead of CFG consisting of basic blocks [6,7]. Usual CFG is a graph with basic blocks as nodes [6,7], on the other hand, CFG here is a graph with ACBs as nodes. A node in the CFG consisting of ACBs represents either a (non-) deterministic assignment or a conditional statement, while an edge in the CFG represents a possible change in the assembly program's control location. Given a transition system $M$, a property $\Phi$, and a bound $k$, Bounded Model Checking unrolls the system $k$ times and translates it into a verification condition $\psi^k$ such that $\psi^k$ is satisfiable if and only if $\Phi$ has a counterexample of depth $k$ or less. The verification condition $\psi^k$ is constructed from the CFG consisting of ACBs, and given by Equation (6).

As $ACB(s^{ACB}{}_q, s^{ACB}{}_{q+1})$ is constructed by composing basic blocks such as $\gamma(s_j, s_{j+1})$, $\gamma(s_{j+1}, s_{j+2}), \ldots$, Equation (6). can compute the larger number of reachable states than Equation (1)

$$\psi^k = I(s^{ACB}{}_0) \wedge$$
$$\bigvee_{i=0}^{k} (\bigwedge_{q=0}^{i-1} ACB(s^{ACB}{}_q, s^{ACB}{}_{q+1}) \wedge \Phi(s^{ACB}{}_i)). \tag{6}$$

Verification property $\Phi$ is intended to be re-given (abstracted by block granularity) in the CFG after blocking with ACB.

An example of merging from two basic blocks to ACB is shown below. Sometimes it merges from two or more basic blocks to the ACB. When the number of instructions of ACB is $L$, $ACB(s^{ACB}{}_q, s^{ACB}{}_{q+1})$ is represented by Equation (7) as shown in Figure 5.

$$ACB(s^{ACB}{}_q, s^{ACB}{}_{q+1}) :=$$
$$G(s^{ACB}{}_q) \wedge IS(s^{ACB}{}_q, s^{ACB}{}_{q+1}). \tag{7}$$

$ACB(s^{ACB}{}_q, s^{ACB}{}_{q+1})$ is the conjunction of a transition condition $G(s^{ACB}{}_q)$ and an instruction constraint $IS(s^{ACB}{}_q, s^{ACB}{}_{q+1})$. An instruction constraint $IS(s^{ACB}{}_q, s^{ACB}{}_{q+1})$ is the logical formula representing effect by executing the instruction.

We suggest thefollowing:

1. When instructions included in one *ACB* are few, the effect of reducing state spaces is not provided enough.
2. As the length of logical formula $ACB(s^{ACB}{}_q, s^{ACB}{}_{q+1})$ is long, verification time is long by SMT solver.

*5.4. Comparison of Usual CFG, CFG of IHER, CFG of ACB*

In this subsection, we compare usual CFG, CFG of IHER, CFG of ACB. In Figures 6–8, we show the correspondence between the assembly program and the CFG with dotted arrows. The CFGs shown in Figures 6–8 are CFGs considering loops, and include arbitrary execution paths as shown in References [6,7,22,23]. Therefore, if this CFG is verified, it can be verified by model checking for an arbitrary execution path.

1. Usual CFG [6,7] is used in the optimization of compiler. Every assembly instruction, an interrupt occurs. Therefore as usual CFG with interruptions is shown in Figure 6, the state space explosion usually occurs.
2. CFG of IHER is developed by B. Schlich et al. [8]. IHER reduces the number of Interrupt Handler executions by blocking Interrupt Handlers at program locations where there is no dependency between certain Interrupt Handlers and the program [8]. Therefore CFG of IHER is shown in Figure 7.

   For example, in Figure 7, since the value of ER1 after executing SUB.L instruction is the same regardless of the presence or absence of the interrupt routine, the transition to the interrupt routine is decreasing because there is no dependency . As explained in Section 4, IHER and our proposed method analyze even such complex dependencies.
3. We propose CFG of ACB in this paper. We propose ACB by extending Basic block of CFG [6,7] using IHER [8] in order to reduce the state spaces. In order to reduce the number of blocks, we propose the method of making the block of codes, at which interruptions do not occur. Therefore CFG of ACB is shown in Figure 8. Here for example, we show $s^{ACB}{}_q$ $(q = 0, 1, 2, \ldots)$, $PC_p$, $ER_p$ $(p = 0, 1, 2, \ldots)$ in Figure 8 as follows:

   (a) $s^{ACB}{}_0 := PC_0 == 0x0000 \wedge PC_1 = 0x0001 \wedge ER0_0 = 0x0001 \wedge PC_2 = 0x0002 \wedge ER1_1 = ER1_0 + ER1_0 \wedge PC_3 = 0x0003 \wedge ER2_0 = 0x0001$

   (b) The following is repeated with a loop count $n = 0, 1, 2, \ldots$ of lines 3–5 of the program:

   　　i. $s^{ACB}{}_{1+n} := PC_{3+5n} == 0x0003 + 5n \wedge PC_{4+5n} = 0x0004 + 5n \wedge ER0_{1+n} = ER0_{0+n} + ER0_{0+n}$

   　　ii. $s^{ACB}{}_{2+n} := PC_{4+5n} == 0x0004 + 5n \wedge PC_{5+5n} = 0x0005 + 5n \wedge ER1_{2+n} = 0x001$

   　　iii. $s^{ACB}{}_{3+n} := PC_{5+5n} == 0x0005 + 5n \wedge PC_{6+5n} = 0x0006 + 5n \wedge CCR_{0+n} = 0x0$

   　　iv. $s^{ACB}{}_{4+n} := PC_{6+5n} == 0x0006 + 5n \wedge PC_{7+5n} = 0x0007 + 5n \wedge ER1_{3+n} = 0x001$

   　　v. $s^{ACB}{}_{5+n} := PC_{7+5n} == 0x0007 + 5n \wedge PC_{8+5n} = 0x0008 + 5n$

   (c) $s^{ACB}{}_{6+n} := PC_{8+5n} == 0x0008 + 5n \wedge PC_{9+5n} = 0x0009 + 5n \wedge ER2_1 = 0x0000 \wedge PC_{10+5n} = 0x0010 + 5n$

   Here, the *PC* on the sixth line of the program is $PC_{9+5n} = 0x0009 + 5n$, and the *PC* on the seventh line is $PC_{10+5n} = 0x0010 + 5n$.

   However, $n(n = 0, 1, 2, 3, \ldots)$ changes depending on $k$ of bounded model check *k*-bound. The larger $k$ is, the larger $n$ is.

From comparison of usual CFG, CFG of IHER and CFG of ACB, our proposed ACB is effective for the number of blocks as shown in Figures 6–8, where we omit the constraints that represent no change.

## 6. Prototype Model Checker

This prototype is a prototype in the sense that it realizes a subset of instructions of assembly program, but the functions of our proposed method are realized. Therefore, experimental evaluation is general.

### 6.1. Verification Example

In this paper, we demonstrate the effectiveness of our proposed method for robots which carried microcomputer H8/3687 [26,27] of Renesas company. In H8/3687 processor, all the general purpose registers are 16 bits wide. When a general register is used as a data register, it can be accessed as a 32-bit, 16-bit, or 8-bit register. On the other hand, Control registers consist of 24-bit *PC*, 8-bit *CCR*. When ER7 (SP) is used as an address register to access the stack area, the operand size should be word or longword.

In this paper, we model values of 16-bit registers and memory assigned address in the 16-bit by Fixed-Size Bit-Vector theory in SMT [18]. We define registers by Fixed-Size Bit-Vector typed variables, and define memory by the function of Fixed-Size Bit-Vector mapping a 16-bit address to a 8-bit value. We construct states by Fixed-Size Bit-Vector typed variables and the function. Fixed-Size Bit-Vector theory consists of functions and predicates of Fixed-Size Bit-Vector. Functions consist of concat, extract, bvadd, bvsub.

### 6.2. Example of Modeling Assembly Instructions

We transform instructions into instruction constraints as shown in Table 1. The notation of the expression follows input form SMT-LIB2lang [20] of SMT-LIB 2.0. In Table 1, we omit implicit behaviors about *PC*, and *CCR*, and we list only logical formulas indicating the behaviors for the operand.

1.  The data transfer instruction *MOV ERs, ERd* is represented by the equation between source register $ERs_j$ and destination register $ERd_{j+1}$.
2.  *ADD.W Rs, Rd* is represented by the equation between an operation result and a destination register using *_extract*, *concat* and *bvadd*.
3.  @ denotes register indirect addressing. *MOV.B RsL @ERd* stores the low 8 bits of a source operand *RsL* into the address that the value in a destination register points. Access to address space (AS) is defined by a function, and the argument of the function is indirect address.

**Table 1.** Logical formulas of H8/3687 asembly instruction(SMT-LIB2 notation).

| Item | Instruction | Operation | Formula |
|:---:|:---:|:---:|:---:|
| 1. | MOV.L ERs, ERd | ERs32 → ERd32 | $(ERd_{p+1} = ERs_p)$ |
| 2. | ADD.W Rs, Rd | Rd16 + Rs16 → Rd16 | $(ERd_{p+1} = (concat((\_extract\ 31\ 16)ERd_p)$ $(bvadd((\_extract\ 15\ 0)ERd_p)((\_extract\ 15\ 0)ERs_p))))$ |
| 3. | MOV.B RsL, @ERd | Rs8 → @ERd | $((AS_{p+1}(\_extract\ 15\ 0)ERd_p)) = ((\_extract\ 15\ 0)ERs_p)$ and $(forall((x(\_BitVec\ 16)))((x = (\_extract\ 15\ 0)ERd_p)$ $or((AS_{p+1}) = (AS_p))))$ |

### 6.3. Configuration of Prototype Model Checker

We show the configuration of prototype model checker in Figure 9. Our prototype model checker consists of analysis of assembly codes, generation of CFG, VC (Verification Condition) builder, verification by SMT solver. Prototype Model Checker is written in Java 7 (7500 lines).

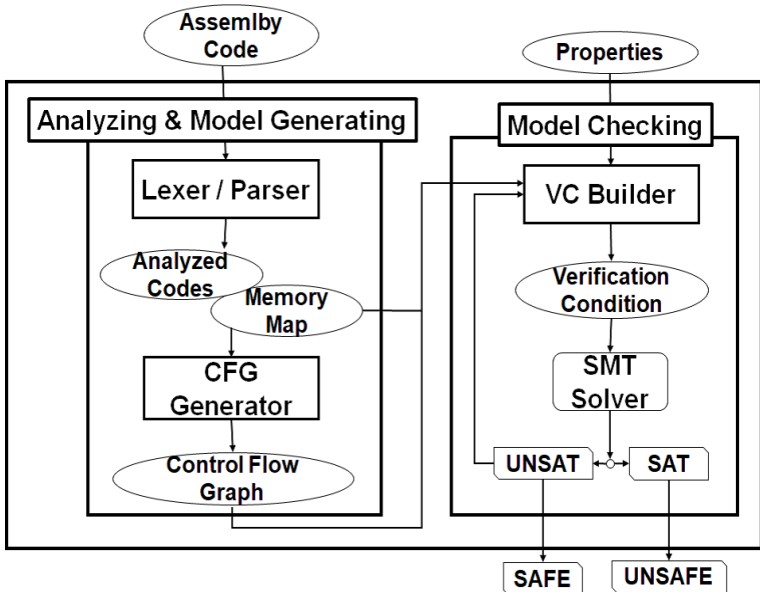

**Figure 9.** Configuration of prototype model checker.

1.  Lexer and Parser:

    Lexer performs lexical analysis, and parser performs syntax analysis of assembly codes of H8/3687 microcomputer. Lexer and Parser output analyzed codes (syntax tree) and memory map from assembly codes. We use JFlex [28] and BYACC/J [29] for Java in order to develop Lexer and Parser.

2.  CFG generator:

    CFG generator constructs CFG from syntax tree. First CFG generator composes interrupts. After that it constructs CFG using IHER. CFG consists of both ACBs and the transitions between ACBs.

3.  VC builder:

    VC builder generates verification conditions. VC builder generates logical formulas of an initial condition and transition relations from CFG. Given a property $\Phi$ and a bound $k$, the verification condition $\psi^k$ is gnerated such as Equation (6).

4.  SMT solver: We verify the verification condition $\psi^k$ using SMT solver Z3 4.3 [30]. In this paper, we use Fixed-Size Bit-Vector theory in SMT [18,24].

## 7. Verification Experiments

### 7.1. Overview of Verification Experiments

Prototype model checker verifies assembly codes of H8/3687 microcomputer. We prepare two programs. Two programs satisfy each verification condition. We verify programs using only IHER and using both IHER and ACB. As SMT solver may not terminate in a realistic time, we stop verification in a time limit.

We verify whether the following property composed of the conjunction of three conditions $\phi_1(s^{ACB}_q)$, $\phi_2(s^{ACB}_q)$ and $\phi_3(s^{ACB}_q)$ is satisfied. Thought the property is a liveness property correctly if infinite k-bound, we verify whether there is a counterexample until finite k-bound. Namely we verify the following liveness property as a safety property.

1.  $\phi_1(s^{ACB}_q)$: Program reaches RTS instruction.
2.  $\phi_2(s^{ACB}_q)$: The value of stack pointer after reaching RTS is equal to the value of stack pointer before executing the routine.
3.  $\phi_3(s^{ACB}_q)$: The value of *PC* after executing RTS is equal to return address.

For verifying the above property, the set of initial states is given by initial conditions as follows.

1. $I_1(s^{ACB}_0)$: The initial value of $PC$ is set.
2. $I_2(s^{ACB}_0)$: The initial value of stack pointer is equal to the value of stack pointer before calling this routine.
3. $I_3(s^{ACB}_0)$: The return address is stored in the stack at the time of the routine start.

In accordance with the above, we give the set of initial states $I(s^{ACB}_o)$ and the verification property $\Phi(s^{ACB}_i)$ in the verification conditional Equation (6) verified in this experiment concretely in SSA form.

1. When the initial value of $PC$ is set to $\sharp x0100$ and the return destination address is set to $\sharp x0116$ in program 1, the set of initial states $I(s^{ACB}_0)$ is defined as follows:

   $I(s^{ACB}_0) = I_1(s^{ACB}_0) \wedge I_2(s^{ACB}_0) \wedge I_3(s^{ACB}_0)$, where $I_1(s^{ACB}_0) = (PC_0 = \sharp x0100)$, $I_2(s^{ACB}_0) = (ER7_0 = \sharp xFF78)$, $I_3(s^{ACB}_0) = (AS_0(ER7_0) = \sharp x0001) \wedge AS_0(ER7_0 + 1) = \sharp x0016)$.

   Here $I_1(s^{ACB}_0)$ is an expression defining the initial value of the program counter $PC$ (corresponding to the initial condition 1), $I_2(s^{ACB}_0)$ is an expression defining that the initial value of stack pointer is equal to the value of stack pointer before calling this routine (corresponding to the initial condition 2), and $I_3(s^{ACB}_0)$ is an expression defining that the return address is stored in the stack at the time of the routine start.

2. When the address of ReTurn from Subroutine (RTS) instruction is $\sharp x010E$, the return destination address is $\sharp x0116$, and the stack pointer register $ER7$ is $\sharp xFF78$ in program 1, the verification property $\Phi(s^{ACB}_q)$ is defined, where $q = 6 + n$ and $p = 11 + 5n$ as follows:

   $\Phi(s^{ACB}_q) = \phi_1(s^{ACB}_q) \wedge \phi_2(s^{ACB}_q) \wedge \phi_3(s^{ACB}_q)$, where $\phi_1(s^{ACB}_q) = (PC_{10+5n} == \sharp x010E)$, $\phi_2(s^{ACB}_q) = (ER7_0 == \sharp xFF78)$, $\phi_3(s^{ACB}_q) = (PC_{11+5n} == \sharp x0116)$.

   As shown in Figure 8, a verifier specify $\Phi(s^{ACB}_q)$ including $q = 6 + n$ and $p = 11 + 5n$ using the correspondence between the assembly program and the CFG with dotted arrows, where $n$ is the number of loops of program.

   Here $\phi_1(s^{ACB}_q)$ is an expression defining that Program reaches RTS instruction (corresponding to the property 1), $\phi_2(s^{ACB}_q)$ is an expression defining that the value of stack pointer after reaching RTS is equal to the value of stack pointer before executing the routine (corresponding to the property 2), and $\phi_3(s^{ACB}_q)$ is an expression defining that the value of $PC$ after executing RTS is equal to return address (corresponding to the property 3).

As a general theory, in embedded software, assembly language description is included in the interface part with hardware, so programming with C program alone is impossible. Therefore, in embedded software, verification of the assembly program is necessary. Also, since one line of the C program is compiled into the assembly program 2–3 lines, it can not be verified accurately by C program verification for interrupt processing occurring for each assembly program line. Also, it can not be verified that it is not an assembly program that the stack is not destroyed. The points concerning the evaluation experiment are as follows.

1. The verification property that the stack pointer before the function call is equal to the stack pointer immediately after returning from the called function can be verified only by the assembly program. By this verification property we verify whether the stack is not destroyed.
2. The verification property that the return address saved before the function call is equal to the address after execution of the RTS instruction of the called function can be verified only by the assembly program. By this verification property we verify whether the stack is not destroyed, too.

*7.2. Which Programs Are Verified ?*

We prepare two programs consisting of main routine and interrupt routines. They are the essential part of typical embedded programs implemented in e-nuvo WHEEL [31].

Program 1 consists of routine *_testCode* and interrupt *_testIntr* as shown in Figure 10. *_testCode* repeats the addition of register *ER*0 until the value of register *ER*1 except 0 is stored. *_testIntr* renews contents of the stack while storing a value in register *ER*1. That is, this program is a program that continues adding until an interrupt occurs and does not stop. Also, assuming that an interrupt is disabled in program 1 in the initial state, the interrupt is enabled by setting a value to the 6th bit in the address indicated by the interrupt status register Inputbuffer Control Register (ICR). As the above verification property is satisfied, model checker outputs UNSATisfiable (UNSAT).We stop verification in a time limit.

Program 2 consists of routine *_sci_txbuf_set* and interrupt *_int_sci*, which are implemented in e-nuvo WHEEL [31] as shown in Figure 11. *_sci_txbuf_set* sets a value in a transfer buffer of serial communication interface (SCI3). This *_sci_txbuf_set* controls an interrupt at the time of the transfer by SCI3, and the transfer processing in interrupt routine *_int_sci* is prohibited during executing this routine. We stop verification in a time limit.

```
_testCode
0:        MOV.L  #1 ,  ER0
1:        SUB.L  ER1,  ER1
2:        BSET.B #6 ,  @ICR
3: LOOP  ADD,L  ER0,  ER0
4:        CMP.L  #0 ,  ER1
5:        BEQ    LOOP
6:        BCLR.B #6 ,  @ICR
7:        RTS

_testIntr
8:        MOV.L  #1 ,  ER1
9:        RTE
```

**Figure 10.** Program 1.

```
_sci_txbuf_set                          _int_sci:
0:     PUSH.W         R6               0:     PUSH.W         R6
1:     MOV.W R7,R6                     1:     MOV.W R7,R6
2:     PUSH.W         R5               2:     PUSH.W         R1
3:     SUBS.L #2,SP                    3:     PUSH.L ER0
4:     MOV.B R0L,R5L                   4:     SUB.W #6,R7
5:     BCLR.B #7,@65450:8              5:     MOV.B @65452:8,R0L
6:     MOV.W @_sci_txend:16,R0         6:     MOV.B R0L,@(-9:16,ER6)
7:     CMP.W #255,R0                   7:     BLD.B  #6,@65450:8
8:     BLT    L57:8                    8:     MOV.B @65450:8,R0L
9:     SUB.W R0,R0                     9:     BPL    L79:8
10:    BRA    L58:8                    10:    MOV.B @(-9:16,ER6),R0L
11: L57: MOV.W @_sci_txend:16,R0      11:    EXTU.W R0
12:    INC.W  #1,R0                    12:    BTST.B #7,R0L
13: L58: MOV.W R0,@(-4:16,ER6)        13:    BEQ    L80:8
14:    MOV.W @_sci_txtop:16,R0        14:    MOV.W @_sci_txcnt:16,R0
15:    MOV.W @(-4:16,ER6),E0          15:    BEQ    L81:8
16:    CMP.W E0,R0                    16:    MOV.W @_sci_txtop:16,R0
17:    BEQ    L59:8                   17:    MOV.B @(_sci_txbuf:16,ER0),R0L
18:    MOV.W @_sci_txend:16,R0        18:    MOV.B R0L,@(-11:16,ER6)
19:    MOV.B                          19:    MOV.B @(-11:16,ER6),R0L
       R5L,@(_sci_txbuf:16,ER0)       20:    MOV.B R0L,@65451:8
20:    MOV.W @(-4:16,ER6),R0          21:    MOV.W @_sci_txtop:16,R0
21:    MOV.W R0,@_sci_txend:16        22:    CMP.W #255,R0
22:    MOV.W @_sci_txcnt:16,R0        23:    BLT    L82:8
23:    INC.W  #1,R0                   24:    SUB.W R0,R0
24:    MOV.W R0,@_sci_txcnt:16        25:    BRA    L83:8
25:    BRA    L60:8                   26: L82: MOV.W @_sci_txtop:16,R0
26: L59: MOV.W #LWORD _sci_stat,R0    27:    INC.W  #1,R0
27:    BSET.B #1,@ER0                 28: L83: MOV.W R0,@_sci_txtop:16
28: L60: BSET.B #7,@65450:8           29:    MOV.W @_sci_txcnt:16,R0
29:    ADDS.L #2,SP                   30:    DEC.W  #1,R0
30:    POP.W  R5                      31:    MOV.W R0,@_sci_txcnt:16
31:    POP.W  R6                      32:    BRA    L84:8
32:    RTS                            33: L81: BCLR.B #7,@65450:8
                                      34: L84:
                                         L80:
                                         L79: ADD.W  #6,R7
                                      35:    POP.L  ER0
                                      36:    POP.W  R1
                                      37:    POP.W  R6
                                      38:    RTE
```

**Figure 11.** Program 2.

### 7.3. Results of Experiments

We verify programs in the following experiment environment as shown in Table 2. We stop verification in ten minutes of SMT solver. If verification time is longer than 3600 s after verification, verification is aborted.

**Table 2.** Environment of verification.

| | |
|---|---|
| CPU | Windows 7 Professional 64 bit |
| OS | Core i7-3770 CPU @3.40 GHz |
| Memory | 16 GB |
| SMT solver | Microsoft Z3 v4.3.0 [30] |
| Java | Ver. 1.7.0_45 |
| Prototype | 8400 lines |

By the introduction of ACB, both the number of interruption codes and the number of blocks are reduced, but the number of codes in a block is increased. The cost of verification is determined by the multiplication of the number of blocks and the size of blocks. Therefore in ACB, the verification cost of each block has a bigger cost. But the overall verification cost decreases because there is little number of blocks. At first, we verify the program with $k = 1$, then we increase $k$ one by one and repeat program verification.

### 7.3.1. Verification Results of Program 1

We show verification results of program 1 using existing IHER and using our proposed ACB as shown in Table 3. When comparing with the same verification time, it is better to use ACB rather than IHER to include more assembly code lines (number of instructions) as shown in Table 3. In Table 3, the column name nodes of IHER is the number of basic blocks and the column name nodes of ACB is the number of ACB blocks. The number of nodes of ACB in the Table 3 was calculated by generating CFG. As $\psi^k$ is unsatisfiable using SMT solver, $\Phi$ is satisfied, and does not have a counterexample of depth $k$.

**Table 3.** Verification result of program 1.

| Model | Nodes | Result | Time (s) | k-Bound | Assembly [Lines] |
|---|---|---|---|---|---|
| IHER | 10 | UNSAT | 3673 | 57 | 3536 |
| ACB | 5 | UNSAT | 4075 | 52 | 4682 |

Also we show verification lines of program 1 for verification time in Figure 12. The vertical axis "Lines" in Figure 12 is the number of lines of the verified assembly code, and it was measured with our verification tool. For all state transition sequences obtained with the number of state transitions less than the given bound, the sum of the number of code lines executed by the program corresponding to each state transition sequence is assembly [lines] in Tables 3 and 4 . When we use ACB, the number of lines which we can verify in 3600 s increases from 3500 lines to 4500 lines. Therefore from the viewpoint of a bound $k$, our proposed ACB is superior to existing IHER. In the verification results of Program 1, both IHER and ACB output UNSAT. As the tool answers UNSAT, the behavior shown by the real microcontroller is safe under general conditions such as initial conditions.

**Table 4.** Verification result of program 2.

| Model | Nodes | Result | Time (s) | k-Bound | Assembly [Lines] |
|---|---|---|---|---|---|
| IHER | 73 | UNSAT | 43,656 | 33 | 4485 |
| ACB | 43 | UNSAT | 43,789 | 22 | 9704 |

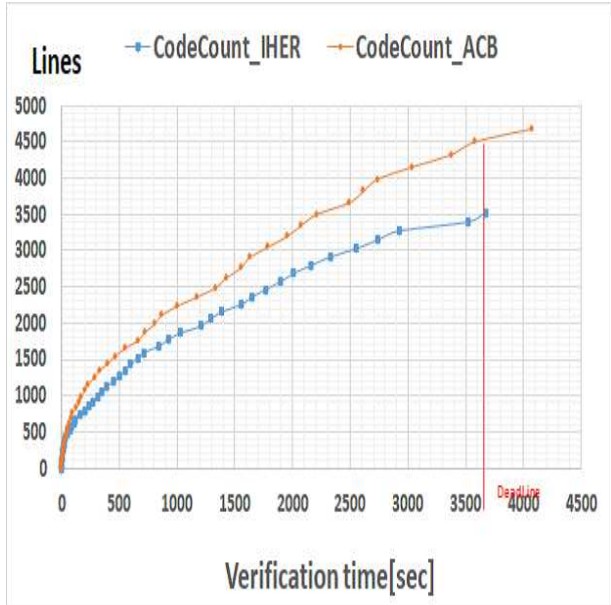

**Figure 12.** Verification lines of program 1 for verification time.

7.3.2. Verification Results of Program 2

We show verification results of program 2 using existing IHER and using our proposed ACB as shown in Table 4. In Table 4, the column name nodes of IHER is the number of basic blocks and the column name nodes of ACB is the number of ACB blocks. When $\psi^k$ is satisfiable using SMT solver, $\Phi$ has a counterexample of depth $k$.

In the verification results of Program 2, both IHER and ACB output UNSAT. Therefore it was confirmed that the stack was used correctly in the execution path that was verified up to each maximum step. Since interrupts are prohibited from line 6 to line 29 of _sci_txbuf_set_ , the reduction of nodes due to blocking was remarkable. Figure 13 is a graph showing the change in the number of verified code lines per verification time. The number of verification lines per second is always higher than that of our ACB model, and the effect of blocking was confirmed, and the efficiency of verification time is considered to be higher for our ACB model.

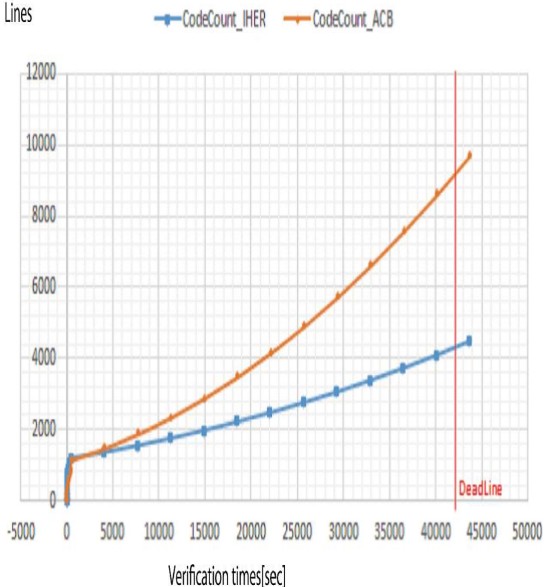

**Figure 13.** Verification lines of program 2 for verification time.

*7.4. Comparison with Classical Model Checking*

7.4.1. Comparison with the Classical Model Checker SPIN

We compare SPIN [32] with our proposed method by program 1. We can verify Program 1 using SPIN in less time than using our proposed method as shown in Figure 14.

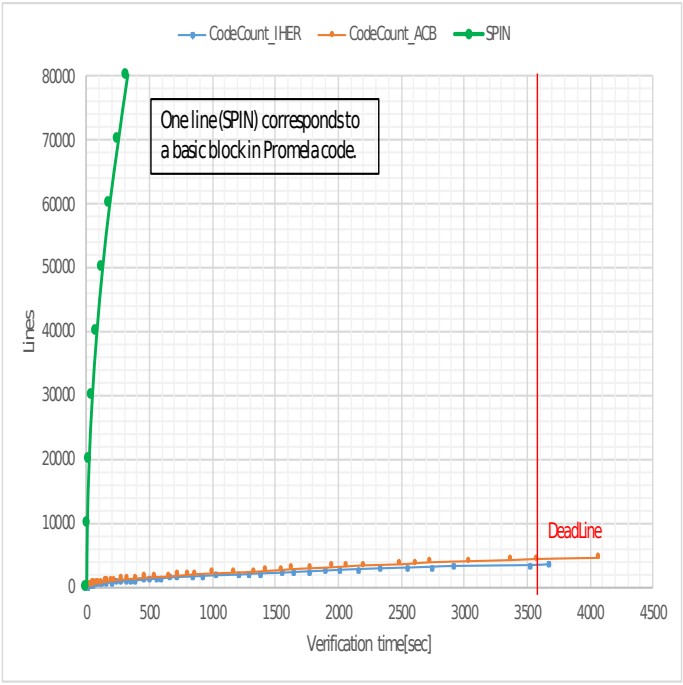

**Figure 14.** Comparison of SPIN and proposed Tool.

We can verify a system in polynomial time for the size of the system using SPIN. Also, SPIN has the following several optimisation algorithms to make verification runs more effective:

1. partial order reduction
2. bitstate hashing
3. minimised automaton encoding of states (not in a hashtable)
4. state vector compression
5. dataflow analysis
6. slicing algorithm

On the other hand, SMT-based model checking is convenient because it can use a general purpose theorem provers, but deciding the satisfiability of formulas with respect to decidable background theories is usually in NP-complete [18].

7.4.2. Examples That Can Not Be Verified by Classical Model Checking

Here we consider Program 3 in Figure 15. As function _$f$ is not defined in Program 3, we can not verify Program 3 by SPIN [32] . On the other hand, we can verify Program 3 using the following SMT theory of undefined function [18] by our proposed method.

$$(x1 = y1) \wedge (x2 = y2) \text{ implies } F(x1, x2) = F(y1, y2).$$

This is one of advantages of SMT-based model checker.

```
_f:
        /* _f is undefined. */
        /* _f receives parameters with ER1 and ER2. */
        /* _f returns the operation result as ER 0. */
        RTS

_main:
/*  Suppose that x1 = y1 and x2 = y2  */
1:  TOP:
2:          MOV.L  #x1,  ER1
3:          MOV.L  #x2,  ER2
4:          BSR _f
5:          MOV.L  ER0, ER3   /* ER3=_f(#x1,#x2) */
6:          MOV.L  #y1,  ER1
7:          MOV.L  #y2,  ER2
8:          BSR _f             /* ER0=_f(#y1,#y2) */
9:          CMP.L  ER3,ER0
10:         BEQ    LAST
11:         BSR    TOP
12: LAST:
13:         RTS

_testIntrB
14:         MOV.W  @(4,ER7), R1
15:         ADD.W  #2, R1
16:         MOV.W  R1, @(4,ER7)
17:         RTE
```

**Figure 15.** Program 3.

## 8. Conclusions and Future Works

In this paper, we proposed the verification method of safety properties using ACB (Assembly Code Block) by combining SMT-Based Bounded Model Checking [9] and Reduction of Interrupt Handler Executions [8]. Also we implemented prototype model checker by Java 7500 lines, and show effective our proposed method. We model registers and values of assembly codes using Fixed-Size Bit-Vector theory, and construct a transition system. Also we construct the transition system including interrupts. We reduce state spaces using ACB.

In this paper, we developed a prototype model checker for demonstrating our proposed methods. We cannot specify verification properties without looking at generated CFG of ACBs. But if we extend our prototype model checker, we can specify verification properties with looking only at assembly program.

We are now extending our prototype model checker for specifying verification properties with looking only at assembly program. We are currently verifying other examples and properties using our proposed method. Also we develop verifying liveness properties.

**Author Contributions:** Conceptualization, S.Y.; Funding acquisition, S.Y.; Methodology, S.Y.; Project administration, S.Y.; Resources, S.Y.; Software, J.K. and K.U.; Supervision, S.Y.; Writing—original draft, S.Y. All authors have read and agreed to the published version of the manuscript.

**Funding:** This work was supported in part by JSPS/MEXT Grant-in-Aid for Scientific Research Numbers 15K00093.

**Conflicts of Interest:** The authors declare no conflict of interest.

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
