# Peer review of "Verification Method of Safety Properties of Embedded Assembly Program by Combining SMT-Based Bounded Model Checking and Reduction of Interrupt Handler Executions"

_electronics, doi:10.3390/electronics9071060_

Round 1
Reviewer 1 Report
The authors develop a verification method for safety properties by combining SMT-Based Bounded Model Checking and Reduction of Interrupt Handler Executions. To show the practical impact of their methology, they test their theory on two examples. In my opinion the paper can be accepted after a minor revision. In particular the authors should clarify in the paper the following two points:
- What is new in the current submission w.r.t. their precedent papers (namely [7] and [8]).
- The methodology proposed seems to be more "precise" than that proposed in [5] but, if the tool answers UNSAT, then the behavior shown by the real microcontroller are all really safe? Or are safe under a (pre)condition? I have lost something?
Author Response
>w.r.t [7] and [8]
This paper is an extention of the previous works in [7] and [8].
>If UNSAT, are thwy all really safe?
If UNSAT, it is safe under general initial conditions.

Reviewer 2 Report
The paper presented the verification method of safety properties of embedded assembly program by combining SMT-based bounded model checking and reduction of interrupt handler executions.
Overall the paper described the method well and experiment is comprehensive. Only I want to see the further discussion with other previous works. The proposed method is only evaluated in some prototype. The general evaluation should be performed.
Author Response
> other previous works?
We add a new reference.
>The general evaluation should be performed.
The prototype is a prototype inthe sence that it realizes a subset of instruction of assembly program, but the functions of our proposed method are realized.
therefore our experimental evaluation is general.

Reviewer 3 Report
GENERAL SEMANTIC ASPECT OF CONTENT
Generally speaking, the paper uses predicate formulas to enable modeling the assembly code in particular use of safety concerns of embedded assembly code. The approach is presented with many technical and theoretical background details (state transitions, predicate logic formulas and use of automated tools). What has not been targeted and explained enough is the precise problem of embedded code (vs. any other assembly code), as well as the more precise safety issues in embedded environment. What aspects of safety were targeted...? Only interrupts and their reductions have some common points. General suggestion is to add related work, theoretical background and existing solutions overview, as well as contributions in empirical sections related to particular concerns - embedded and safety.
GENERAL WRITING STYLE
Paper statements in lines from 39 to 55 could not be treated as general truth and wide-accepted. For example, it is very obvious that these sentences are not precise: "The assembly code is the outcome at the end of the development process. Hence, all errors introduced during the complete development process can possibly be found." Instead of this, it is closer to truth that "all errors...are consequently included in assembly code outcome, but by analyzing only assembly code it is not possible to find and detect the roots of the assembly outcome...
Authors should avoid personal attitudes to be included in text, such as "This makes assembly constructs easier to handle than certain C constructs such as pointer arithmetic or function calls via pointers." 2nd example: "It is a tedious work."
Authors should explain basic terms, before their first use. One such term is "model checking assembly code". It has been used in "When model checking assembly code, the model checker does not have to exploit the compiler
50 behavior, hardware-dependent constructs can be handled". It has been explained later in "using SMT solvers we can properly model software, and verify it".
At first use of abbreviation, there should be the whole words explanation next to their occurrence. Example: there is in abstract ("SMT") which has not been explained before.
FORMATTING AND ENGLISH LANGUAGE ISSUES
Authors should be more careful in writing English sentences, since sometimes they could not be understood properly, such as in "Therefore using SMT solvers we can properly model software, and verify it". Writing style is sometimes inappropriate. Spelling is also problematic, e.g. "arithmetics" -> "arithmetic"...
Authors should be careful with formatting, e.g. table at page 16 goes out of margins.
Some words are not appropriately written, such as "assemble code".
Title Case is expected at some spots, starting with author names ("k" - kosuke Uemura).
Author Response
>What has not been targeted and explain enough is the precise of embedded code.
We explain what has been targeted, and explain the precise of embedded code in introduction and related works.
Also we improve general writing style and formating.
